# Qualitative investigation of relatives' and service users' experience of mental healthcare for suicidal behaviour in bipolar disorder

Caroline Clements ![ORCID],[1] Navneet Kapur,[1] Steven H Jones,[2] Richard Morriss,[3] Sarah Peters[4]

[1]Centre for Mental Health and Safety, University of Manchester Faculty of Biology, Medicine and Health, Manchester, UK
[2]Health Research, Lancaster University Faculty of Health and Medicine, Lancaster, UK
[3]Psychiatry, University of Nottingham, Nottingham, UK
[4]Psychology and Mental Health, University of Manchester Faculty of Biology, Medicine and Health, Manchester, UK

**Correspondence to**
Dr Caroline Clements;
caroline.v.clements@manchester.ac.uk

## ABSTRACT

**Objective** People with bipolar disorder are known to be at high risk of engaging in suicidal behaviours, and those who die by suicide have often been in recent contact with mental health services. The objective of this study was to explore suicidal behaviour in bipolar disorder and how this is monitored and managed by mental health services.

**Aims** To identify themes within relatives' and service users' accounts of mental healthcare, related to management and prevention of suicidal behaviour in bipolar disorder.

**Design** Thematic analysis of 22 semistructured interviews.

**Participants** Participants were aged 18 years or over, fluent in written and spoken English, and either had bipolar disorder with a history of suicidal behaviour, or were relatives of people with bipolar disorder who had died by suicide.

**Setting** England, UK.

**Primary outcome** Themes identified from participants' accounts of mental healthcare for suicidal behaviours in bipolar disorder.

**Results** Two main themes were identified. 'Access to care' was characterised by a series or cycle of potential barriers to care (eg, gate-keepers, lack of an accurate diagnosis) which had the potential to increase risk of suicidal behaviour if failure to access care continued over time. 'Problems with communication' captured the importance of maintaining open routes of communication between all parties involved in care to ensure successful monitoring and management of suicidal behaviours in bipolar disorder.

**Conclusions** Mental health services need to be accessible and respond rapidly to people with suicidal behaviour in bipolar disorder. Open communication and inclusion of relatives in care, where appropriate, could help closer monitoring of changes in symptoms that indicate increased risk.

## Strengths and limitations of this study

► This was a novel qualitative exploration of suicidal behaviour in bipolar disorder.
► Thematic analysis of in-depth semistructured interviews generated rich and complex participant-driven data that offered new insight into why suicidal risk in people with bipolar disorder might be missed by mental health services.
► This study is exploratory in nature, and further work is needed to confirm which themes/subthemes are specific to people with bipolar disorder.

(such as employment, social, and physical health), and are known to be at high risk of engaging in suicidal behaviours.[2 3] More than half of people diagnosed with bipolar disorder self-harm at least once in their lifetime, and self-harm is known to be a major risk factor for future suicide.[4 5] Indeed, risk of suicide in people with bipolar disorder has been estimated to be 20–30 times that of the general population.[3 6–9] Some authors suggest that failure to identify elevated risk of suicidal behaviours in people with bipolar disorder reflects the nature of the disorder, with rapid changes in mood-state alongside impulsive characteristics, leading to limitations in assessing diagnosis-specific risk.[10 11]

A UK study of people with bipolar disorder who died by suicide while under the care of mental health services found that 60% had been seen by the care team within 7 days of the suicide, and 40% were seen within 24 hours.[12] People with bipolar disorder who present to hospital following self-harm often report being currently under the care of community mental health services.[13] Such results suggest that risk in people with bipolar disorder is not being appropriately identified or responded to and therefore opportunities for intervention are being missed.

## INTRODUCTION

Bipolar disorder is a common mental illness with an estimated prevalence over 2.4% worldwide.[1] People with bipolar disorder are subject to potential burdens and reduced quality of life across a number of domains

Treatments shown to be effective at reducing suicidal behaviour, such as cognitive–behavioural therapies or medication with lithium are perhaps not being successfully implemented by mental health services in response to suicidal crises in people with bipolar disorder.[14–18] However, work with people with other psychiatric diagnoses and general population samples have revealed common difficulties in access to and/or engagement with mental health services in relation to suicidal behaviours.[19]

A number of studies have investigated barriers and facilitators of access to/engagement with mental health and medical services in general service user groups,[20–22] people with psychiatric diagnoses[19] and after episodes of suicidal behaviour.[23–25] Qualitative methods have been used to explore the broader experiences of people with bipolar disorder (and their carers), such as social factors in suicidality,[26] psychiatric assessment[27] and other aspects of mental healthcare.[28–30] A recent study by Vallarino *et al* explored the experiences of mental healthcare in people with bipolar disorder; timely provision of information about bipolar disorder, recovery-focused treatment, provision of psychosocial therapies and access to peer support, all had an impact on attitudes to and level of engagement with care services.[31] To the best of our knowledge, there has been no investigation specific to accessing care and support for suicidal behaviours in people with bipolar disorder. Understanding the experiences of people with bipolar disorder with regard to how mental health services respond at times of crisis and increased risk may be key to reducing suicidal behaviours. The aim of this study was to explore relatives' and service users' experience of mental health services in relation to suicidal behaviour in bipolar disorder and to look at why suicidal risk in this population might be missed.

## METHOD
### Participants
Participants were: (1) relatives/carers of people with bipolar disorder who died by suicide (hereafter referred to as 'relatives'), or (2) people with bipolar disorder who had a history of self-harm (hereafter referred to as 'service users': self-harm is defined here as any self-poisoning or self-injury without reference to motivation or level of suicidal intent[32]). To capture the broadest range of experiences possible inclusion criteria were limited to being over 18 years of age and fluent in written and spoken English (use of translators was beyond the scope of this study and fluency in English was essential enable fully informed consent). To maximise recruitment from this small and highly specific target population no time-limits were placed on when the suicidal behaviour had taken place.

Relatives and service users were included in order to encompass the full range of suicidal behaviour in bipolar disorder. While people with bipolar disorder who have experience of self-harm can speak to their own experiences, excluding information about those who died by

suicide may neglect the experiences of a group with the highest need for intervention by mental health services. People bereaved by suicide are often included as informants in suicide research as they are well positioned to know details of the individual and circumstances prior to death that are unavailable from clinical records.[33 34] Although some significant events may be unknown to the bereaved person (in addition to being subject to recall bias and the influence of distress on the overemphasis or underemphasis of salient factors) the inclusion of relatives is recognised as a pragmatic approach.

Given the sensitivity of the topic and potential for burden and distress, no formal assessments were conducted to establish a diagnosis of bipolar disorder. Inclusion was based on self-reported diagnosis or diagnosis as reported by relatives. Interviews did cover diagnostic history, and it was possible to confirm that 19 (all 11 service users and most who died by suicide) had received a formal diagnosis of bipolar disorder, while 3 had a provisional or suspected diagnosis made via a mental health professional prior to death. Service users' experience of self-harm included various behaviours, from medically minor self-injury to medically serious suicide attempts. Changes in method of self-harm, severity, and intent over time were common, which is consistent with other studies of self-harm.[35]

### Recruitment
Twenty-two people were recruited into the study, split equally between relatives and service users.

Recruiting from such a small and highly specific population was challenging and a variety of pragmatic methods were used. Recruitment initially took place in the North West area of England, but by necessity was expanded to include all of England (amendments were approved by the National Research Ethics Service to reflect these changes). Identification of potential participants was done using the following methods; identification of people who had presented to hospital following self-harm via local psychiatric liaison services in the City of Manchester; identification and referral via clinical studies officers based within the local National Health Service (NHS) mental health trust; advertising via mental health charities and local newspapers; in-person visits to local voluntary sector support groups, for example, Bipolar UK; and personal referrals from other participants and research colleagues.

Potential participants were given a detailed participant information sheet and an opportunity to discuss the study further with the research team, before giving written informed consent to participate in the interview. Details of the study and procedure were repeated at the time of the interview and consent reconfirmed verbally.

### Interviews
Semistructured interviews were conducted using a topic guide to facilitate conversation. The topic guide was based on factors relevant to suicidal behaviour in bipolar disorder identified from existing research literature.

Topics included history of diagnosis, changes in mood or behaviour immediately before the most recent suicidal behaviour, situational factors such as negative life events or stressors, experience of mental healthcare, social support and whether opportunities to prevent the suicidal behaviour had been missed. As this work was exploratory in nature, participants were encouraged to tell the story of their experiences in their own way and encouraged to raise any additional issues they felt were important. Two authors (SP and CC) developed the interview procedure with input from the other authors, and one (CC) conducted the interviews. An outline of the sorts of topics that might be discussed was presented at the start of the interview to ensure that the participant was comfortable with the topics, and the semistructured nature of the interview was explained to ensure that participants knew they were in control of the conversation and could discuss any topics they thought important to the experience of suicidal behaviour in bipolar disorder.

Participants were interviewed in-person at the participant's home (n=10), or by telephone (n=12), ensuring the participant had appropriate levels of comfort and privacy in which to talk. Interviews were audio-recorded and transcribed verbatim. A code was allocated to each interview and identifying information (eg, names and places) was removed to maintain anonymity. Interview duration ranged from 30 to 90 min, with a mean duration of 1 hour. The use of both interview methods was a pragmatic decision to aid recruitment and thereby access a range of perspectives and experiences. There are potential advantages and disadvantages to using both telephone and face-to-face interviews. Interviews via telephone can add distance and detachment during a potentially sensitive interview and help people feel more comfortable disclosing personal information[36 37]; however, face-to-face methods are known to increase rapport in the interview setting and may help put participants at ease.[38 39] There was no discernible differences identified between the telephone and face-to-face interviews in topics covered or themes identified in the analysis.

### Service user and public involvement

This work was carried out as part of the Psychoeducation, Anxiety, Relapse, Advance Directive Evaluation and Suicidality (PARADES) programme of research into bipolar disorder. A service user reference group for the programme had input into the development of the project from an early stage. Methodology and recruitment materials were reviewed by the group and amendments made in line with feedback. A first draft of the topic guide was confirmed to be generally suitable for the purpose of the project, but the group advised that more detail be collected on past history of self-harm and interaction with services prior to the most recent episode of suicidal behaviour to give additional context to participants' experiences. The interview was also piloted with a service user who confirmed the interview procedure and topic guide were comprehensive and appropriate for the research topic.

### Analysis

Thematic analysis was used on the interview data.[40] As this method is unattached to a specific theoretical framework and offers flexibility to explore and interpret detailed qualitative data, it is particularly suited to pragmatic and exploratory work.

Detailed information on the analytic procedure is presented in online supplementary table S1. To summarise, initial coding was conducted by CC, transcripts were read by CC and SP to confirm concordance of coding and ideas during analysis. Data were given preliminary codes which were organised within a text document. An iterative process of discussion and review between CC and SP consolidated codes into potential themes (these codes were grouped together in the working document). Themes were summarised and discussed with the extended research team to refine final themes. Conceptual models depicting possible relationships between subthemes were developed during analysis to aid understanding of each themes and how subthemes may be linked. The typical progression or sequence of events over time as described by participants were noted and used to help construct these models which were reviewed and amended by the research team as analysis progressed.

Steps taken to ensure the trustworthiness of the analysis included a multidisciplinary team that comprised qualitative methodologists, clinical psychologists, psychiatrists and expert suicide and self-harm researchers. Throughout the study the team were reflexive, using individual reflexive notes and group discussions. This involved considering their own personal experiences and positions and how these may impact on the data generation, interpretation and final analysis. In doing so, the researchers did not aspire to objectivity but to understand the subjectivity inherent in the process of qualitative research.[41]

### RESULTS

A summary of participants' age and gender are provided in table 1.

| Table 1 Gender and age of interview participants | |
| --- | --- |
| Variable | Participants n=22 (%) |
| Sex | |
| Male | 4 (18.2) |
| Female | 18 (81.8) |
| Age group | |
| 18–35 | 4 (18.2) |
| 36–50 | 10 (45.5) |
| 51–65 | 4 (18.2) |
| 66> | 4 (18.2) |

Analysis of interview data identified two main themes: 'access to care' and 'problems with communication'. Themes and subthemes are presented below. For the purpose of brevity abridged quotes (presented in *italics*) by relatives [r] and service users [s] are presented. Full quotations and additional supporting quotes are provided in online supplementary table S2 and online supplementary table S3.

## Access to care

Participants reported a range of difficulties when trying access to mental health services when risk of engaging in suicidal behaviour was high. This applied across different service-contact situations, such as outpatient care, people discharged or disengaged from services, as well as people trying to access care for the first time. Within this, the subthemes 'help-seeking', 'knowing how to navigate the system', 'gatekeepers', 'obtaining a correct diagnosis' and 'increased risk due to being denied care' formed a series of potential barriers to accessing successful mental healthcare for suicidal behaviours in bipolar disorder. While barriers could be experienced as one off events, many accounts reported a sequence of events where one barrier led to the next and so on, until access to services was achieved.

*Help-seeking:* Most accounts showed that efforts were made to actively seek help for suicidal behaviours from health/mental health services, particularly at times when suicidal thoughts or behaviours increased.

He'd put himself in hospital, he hadn't been sectioned or anything, because he just felt so suicidal and depressed [r1].

*Knowing how to navigate the system:* Participants reported difficulties navigating access to appropriate mental health services. There was confusion over what services were available which would provide the right support for people experiencing suicidal thoughts and behaviours, and how to initiate contact based on current patient-status (eg, service provision differed depending on whether they were a current, discharged or new patient). At times of suicidal crisis, when urgent help was needed, knowing how/who to contact for effective support was seen as vital—however, services did not always respond promptly.

We didn't realise that we had a pass [...] back into the mental health system so we went back to the GP (General Practitioner) [r6].

I went to see the doctor and she said [...] I am going to get someone to ring you from the crisis team [...]. Do you know how long it took? They never phoned me five weeks. I suffered like that for five weeks. [s7].

Confusion over what services was not restricted to service users/relatives, other medical professionals such as GPs also struggled when navigating the mental healthcare system.

[T]he GP had phoned up the psychiatrist's secretary as well, and had got exactly the same. It was ring five numbers on a piece of paper, she said phone them so I phoned them, and they gave me another number so I phoned them [...] and it went round in a circle, so in the end the fifth person gave me the number of the first person [...] and my GP got given the same ring of numbers and went round the same way [s5].

There were also difficulties experienced when trying to access additional support services (unavailable on the NHS or only available via long waiting-lists such as talking therapies). Where participants were willing and able (eg, financially) to tap into additional sources of mental health support, there was a wish expressed for guidance and signposting by the existing care-team. This type of guidance was either limited or refused, leaving service users/relatives frustrated.

I said can I short circuit it by paying for it and she sort of said 'I'm not allowed to recommend anyone.' So it was this, I'm trying to do the best for [the service user] and you won't help me [r2].

*Gate-keepers:* The first point of contact during times of increased risk of suicidal behaviour was often the GP, who then acted as a gate-keeper for accessing secondary mental health services. In this role, GPs could be facilitators or barriers to access.

I cannot fault my GP, because they [...] could not have dealt with it any better [r3].

Gate-keepers' poor attitudes and/or lack of knowledge about how to respond to people with bipolar disorder experiencing suicidal thoughts and behaviours were a common feature in participants' accounts.

I have to say the GPs were very much, 'oh pull yourself together,' you know, sort of attitude [r6].

Hospital emergency departments were another critical point of contact and gate-keeping of access to mental healthcare. Participants were often subject to the negative attitudes of medical staff responsible for their treatment, and for referring those who present with suicidal behaviours to psychiatric teams. Psychiatric staff were also seen as gate-keepers of access to appropriate care, demonstrated by failures to recognise bipolar disorder symptomatology and the associated increased risk of suicidal behaviours in this population.

I was as honest as I am being with you, and [psychiatric liaison clinician] just turned to [psychiatric liaison nurse] and he went, 'do you believe a word she's saying?' And this woman just looked at him and went, 'no' and I felt about that big, and I'm sort of thinking I need help… help [s1].

While the quotes above describe first attempts to access mental healthcare, gate-keepers were also a potential barrier to ongoing successful care for those already

in contact with secondary mental health services. For example, when the psychiatrist responsible for a service user's care is unavailable at a time of crisis:

> [S]o there was another psychiatrist at the hospital, they wouldn't see me because I was under the care of [consultant psychiatrist]. So they wouldn't touch me [s5].

*Obtaining a correct diagnosis:* A common experience among participants was the failure of clinicians to recognise a diagnosis of bipolar disorder. By failing to take account of the individual's diagnosis clinicians failed to correctly evaluate increased risk of suicidal behaviour. Lack of a formal diagnosis also impacted on access to care and increased the risk of inappropriate or inadequate medication for those already in contact with services, with the potential to exacerbated symptoms and increase suicidal behaviours.

> [T]hey put me on antidepressants and I went really, really manic for seven months or something […]. So I was just left on these antidepressants going higher and higher [s5].

*Increased risk due to being denied care:* The final subtheme represents the vulnerability of participants to the emotional and illness-related toll of repeated failures to access care (eg, when failing to overcome the barriers presented in previous subthemes). Participants were clear that this had potential to become a cyclic pattern of repeated attempts and failures to access care, with the attendant disappointment and frustration precipitating further suicidal thoughts and behaviours; leaving participants feeling like they have to 'scream for help [s2]'.

> The crisis team, the home treatment team, nobody would see me, and that is, that is when this [suicidal behaviour] happened, really I was just really desperate [s5].

There was also distress at often not being seen as 'ill enough' to deserve mental healthcare. This was particularly evident when service users looked well or were able to articulate concerns about their illness and the risk of engaging in suicidal behaviours (it is worth noting the interview sample in this study generally had a high level of educational attainment which may be related to the ability to better describe symptoms and distress).

> I find with services they get quite confused you know they look at me and say, 'You are not depressed, you are bathed and clothed.' I am terribly depressed even though I am bathed and clothed [s3].

Barriers to accessing care were evident throughout the data, but a minority of participants reported positive experiences whereby access to care for the prevention of suicidal behaviours was facilitated by gatekeepers and/or knowledge of which service to contact in a crisis. Therefore, barriers discussed here only have the potential to interrupt or block access to care.

A model of relationships between subthemes is provided in figure 1 to aid in interpretation of the 'access to care' theme. 'Help-seeking' is represented as the first step in the model. This connects directly to issues around 'knowing how to navigate the system', as correct knowledge may facilitate access to care but it may also act as a barrier. This subtheme is then linked to 'gatekeepers' and on to 'obtaining a correct diagnosis'. Another failure at this point leads into the theme of 'increased risk due to being denied care'. The circular path of the arrows, denoting relationships between subthemes, represents the potential for individuals to become trapped in a cycle, this is only one interpretation and barriers may be encountered in any order and access to care maybe achieved at any point.

### Problems with communication

This theme describes the importance of clear communication in the ongoing care of people with bipolar disorder, especially at times of crisis when suicidal behaviour is likely. Paths of communication are between any parties with an interest in the care of a service user, and are here presented in relation to the 'service user', 'relatives' and the 'mental health team'.

*Service users:* Service users were the main source of information about changes in illness, behaviour and risk of suicidal behaviour. Participants acknowledged that suicidal intent or increases in suicidal thoughts and behaviour were not always disclosed. As a consequence carers and services remained awareness of changes in symptoms and behaviours that might indicate increased risk (eg, non-fatal suicidal behaviour) remained hidden, and opportunities to intervene were lost.

> I didn't want a professional to know. It's classic hiding it and not telling anybody, because they have to intervene [s2].

Failure to disclosure information that might help manage suicidal behaviour was not always intentional on behalf of the service user. Participants gave examples of memory problems and disorganised lifestyles (eg, substance use problems or lack of a permanent address) as possible reasons why important changes may not be reported.

> I used to get quite cross […], he would say he hadn't seen [his community psychiatric nurse], but when I inquired into it, the more likely thing was that they had gone to his house and he wasn't there [r8].

Motivation for failing to communicate suicidal thoughts and behaviours was driven largely by the wish to protect others from distress and to avoid placing additional burden on relatives.

> [S]he didn't want to worry us [r7].

> I keep a lot to myself and I don't know why I just, I just feel as though if I say things I will upset people [s6].

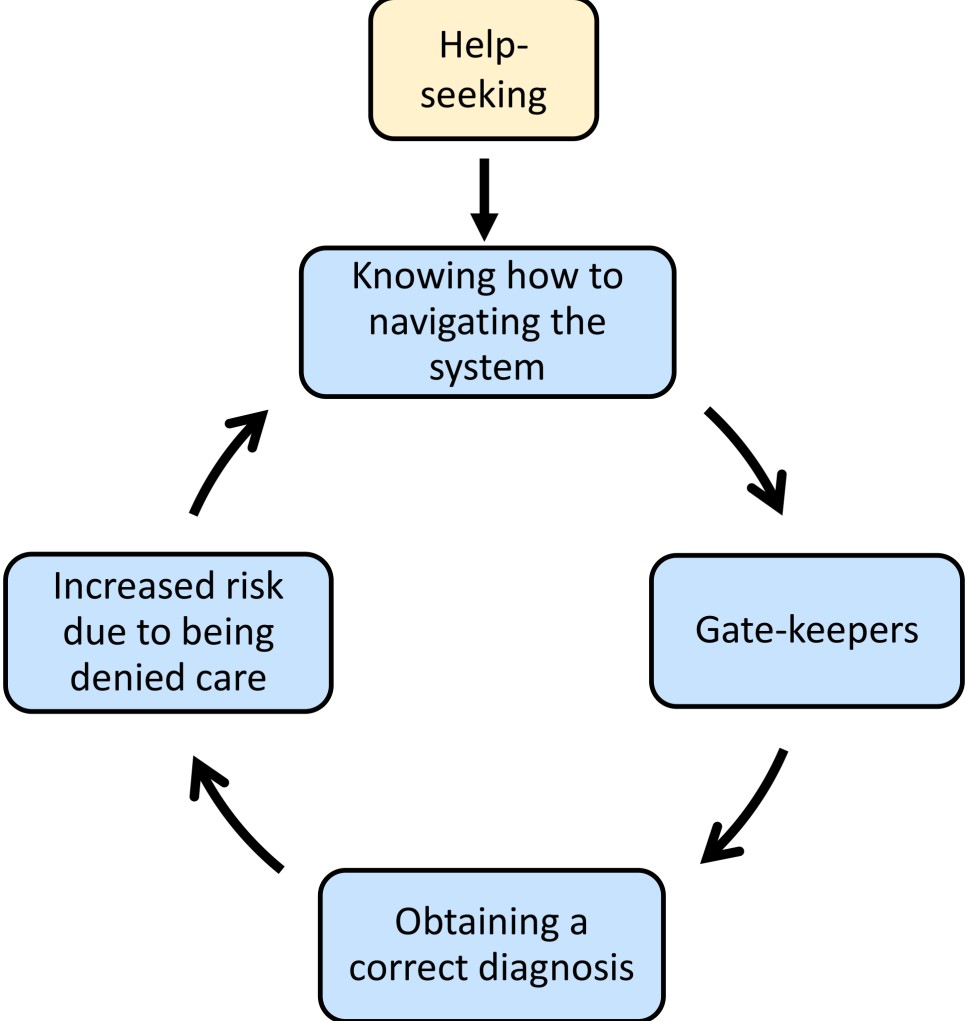

**Figure 1** Model of subthemes within the main theme 'access to care'.

The tangible consequence of this was that relatives and/or the mental health team were unaware of increasing risk and unable to respond effectively.

I came down and found him in the chair. He took an overdose and he'd promised [the psychiatrist] that he wasn't suicidal. [r1].

*Relatives:* The involvement of relatives in mental healthcare was generally viewed as positive. Proximity to the service user helped with observation of changes in illness and behaviour. Relatives were also privy to changes in situational factors, such as the presence of stressful or negative life events that could increase the risk of suicidal behaviour, and which could then be communicated to the care team.

[W]e were aware that he was drinking. […] I brought all this to the attention of his CPN […] and he was quite shocked by that, and the living conditions, how [the service user] was living [r4].

Relatives saw themselves as an untapped resource for services, and wanted to be considered and treated, as full partners in the care of the service user.

[A]s a family we were their biggest resource and they never sort of came to us to help. Either to help us, or to use what we were saying to help him [r4].

Not all service users felt supported by relatives or wanted the support of relatives in the management of their illness or suicidal behaviours. For a couple of participants, there was resistance on the part of the family to being involved in care. In these cases a lack of knowledge of bipolar disorder and the associated risks of suicidal behaviour were cited as obstacles to engaging family members in the care of the service user. Where family involvement was wanted, mental health services were viewed as a source of engagement and education for relatives, and there was a view that working collaboratively would improve support and safety for the service user at home.

I would like somebody to come here and sit them all down and tell them what it involves because I don't think they have a clue [s7].

Where familial relationships were poor the involvement of family members was seen as potentially detrimental to the service user and their ongoing care.

[T]the services would be so grateful to have family involved that they push the care to the family where really that is detrimental to me [s3]

Participants also gave accounts of conflict between family members and mental health staff. This was particularly evident when relatives felt prevented from contributing to care—which often went hand-in-hand with confusion over confidentiality rules.

You just got told on many, many occasions that, you know, its patient confidentiality. […] I turn round and said to them, I said "if my son had cancer, you would have involved me, to care for him quite well at home [r4].

This led to failures in communicating information that had potential value for assessing risk. Relatives expressed frustration at being unable to help the service user despite being in possession of knowledge that may have influenced care.

[W]hen he died, you know, the psychiatrist had actually said that he didn't think it was planned suicide, […] even though he had been telling me that he wanted, you know, he didn't want to live. He wanted to die for probably months [r6].

*Mental health teams:* Participants gave multiple accounts of communication failures between service users and the care team. Poor explanations of diagnosis and prognosis, risks and triggers, medication regimens and possible side effects of treatments, all contributed to service users feeling unheard by clinical teams. This resulted in poorer

management, weaker therapeutic relationships and ultimately the potential to miss indicators of increased risk.

[T]hey never said to me, you know your mom has mental illness you might need to you know go and get yourself checked out at the doctors, with your suicidal thoughts, suicidal tendencies… [s4].

Following episodes of suicidal behaviour, some participants reported being left without any follow-up or acknowledgement from clinical teams. Subsequently, there was no adjustment of treatment, additional interventions or changes in monitoring in response to the suicidal behaviour.

[O]ne of the home treatment team came into the appointment and sat in the room with me, and neither she nor the psychiatrist mentioned the fact that I had self-harmed and run away [s5].

There was however evidence that communication between services and service users has improved over time. With accounts of new teams and initiatives created to better ensure the safety of service users, and overall participants were hopeful that services would continue to improve.

[S]he came out on the Friday and brought the crisis team out […] they said if I needed them over the weekend that they could you know, I could ring them and they would come and see me. So you have a bigger support work now than you did a few years ago [s4].

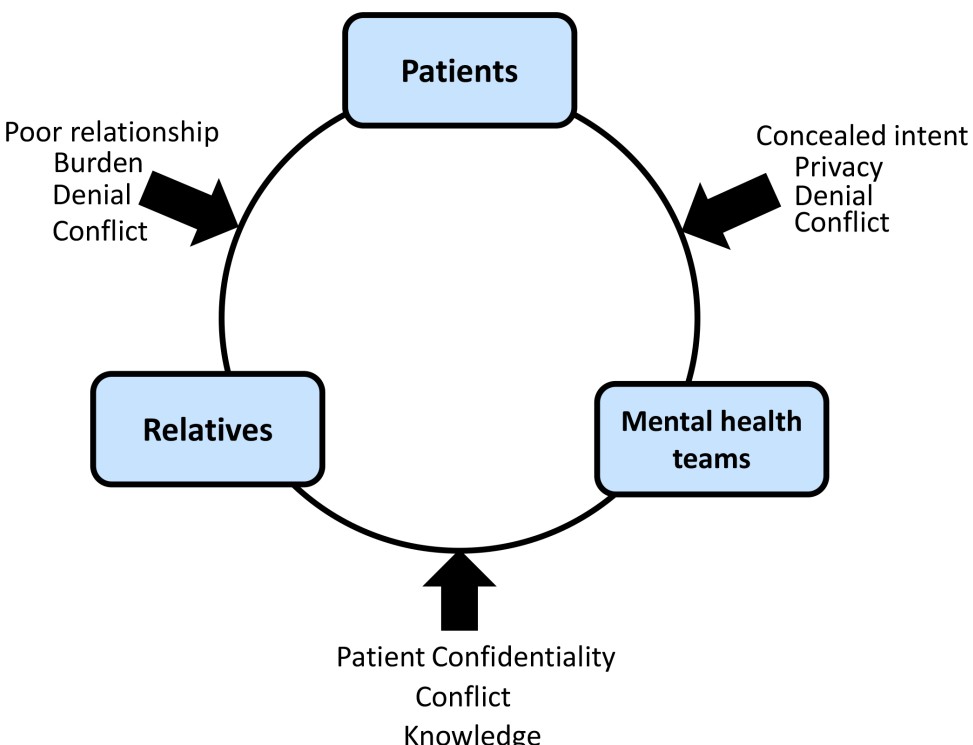

**Figure 2** Model of subthemes within the main theme 'problems with communication'.

The theme 'problems with communication' encapsulates many of the vulnerabilities in routes of communication between service users, relatives and mental health services, in the provision of care for people with bipolar disorder and suicidal behaviour. It is interesting to note that the converse of this was also true; participants' judgements of the quality of care received were more positive in cases where the care team were seen as collaborative and inclusive. Where communication between parties was good, opinions on the quality of care received were positive regardless of outcome.

> [T]here was one guy, he was one of the few [health care professionals] that both [the service user] and I trusted because he was quite happy to have us both in at the same time [r2].

Sympathy and trust were key features of positive evaluations of care. Clinicians who were willing to provide and receive information from service users and relatives were valued. Continuity of care to establish long-term therapeutic relationships were also vital for building trust, and allowed clinicians greater understanding of service users' illness, behaviour and situation; aiding monitoring of bipolar disorder symptoms and identification of times of increased risk of suicidal behaviour. Where these factors were lacking, participants evaluated care as poor.

> I gave up [on] the psychiatrist because they were a bloody waste of time, they didn't know me. […] They just read the notes and there was no connection at all [s7].

A model of subthemes and vulnerabilities in communication is provided in figure 2 to aid interpretation of the 'problem with communication' theme. 'Service users,' 'relatives' and 'mental health teams' are connected to each other and communication can flow in any direction. When communication is open and positive care is viewed as more collaborative and satisfaction with the care team is higher. Examples of problems in communication are given outside the pathways of communication with arrows denoting the possibility that any of these issues could interrupt communication at any time, and impact on the care of the service user.

## DISCUSSION

This exploratory study used a qualitative approach to investigate experiences of mental healthcare in relation to suicidal behaviour in bipolar disorder. Two main themes captured the focus of participants' accounts—access to care and problems in communication—both of which may have contributed to failures to identify and respond to increased risk of suicide in bipolar disorder.

A number of subthemes identified were consistent with experiences common among people with other psychiatric diagnoses.[21 22] While this does not diminish the importance of these concerns, their impact on the individual, or the possibility these difficulties may be over-represented among people with bipolar disorder, the focus here is on the identification of novel factors that may be specific to care for suicidal behaviours in bipolar disorder.

Active help-seeking was common with the majority of participants recognising the need for increased support when experiencing increased suicidal thoughts and behaviours. Barriers were predominantly external, such as gate-keeping and poor staff attitudes to suicidal behaviours.[42 43] There was no evidence of issues relating to stigma or other individual-centred barriers as documented in more general studies of people with bipolar disorder.[21 22] However, it is possible that internalised stigma may have played a role in the decision not to disclose suicidal thoughts to relatives or care teams, which would fall under the theme 'problems with communication'.[44]

One aspect of gate-keeping that may be particularly relevant people with bipolar disorder was the assumption of low illness severity (and therefore low risk of suicidal behaviours) based on superficial aspects of participants' appearance and the ability to effectively communicate concerns and distress. Bipolar disorder is characterised by periods of illness and relative stability when individuals may be better able to effectively recognise and communicate distress and risk.[3] Compared with people with other psychiatric diagnoses, people with bipolar disorder often have a higher educational attainment level, similar to that of the general population.[45 46] Although it was not included as a subtheme, the participants in the current study were particularly well educated—many at degree level or higher—perhaps suggesting this sample had more effective communication skills, especially when describing risk of suicidal behaviours.

Ensuring accurate diagnosis was critical in the management of suicidal behaviours in bipolar disorder. Compared with other psychiatric diagnoses, a diagnosis of bipolar disorder is often delayed due to misdiagnosis of initial symptoms especially when symptoms of depression are present.[47] The experiences of participants who received inappropriate treatments and the exacerbated symptoms and increased risk of suicidal behaviours that followed, emphasised the importance of increasing clinician knowledge, especially for those who may be a first point of contact such as GPs and psychiatric liaison staff, in recognising bipolar disorder and the risks associated with this illness.

Mental healthcare for suicidal behaviour was enhanced by involving service users and relatives (where appropriate) in care decisions and resulted in positive evaluations of care even after a death by suicide. Work by Owen *et al* on the impact of social factors in suicidal thoughts showed that 'not being understood or acknowledged' and 'feeling burdensome' triggered or exacerbated suicidal thoughts and acts.[26] Participants' accounts of problems in communication in the current work echo these results, especially in terms of non-disclosure of suicidal intent to avoid burden on family members. However, participants often wanted the support of family, and the involvement of family in care decisions was highly valued. Therapies that involve family members (eg, family-focused therapy, psychoeducation) have been shown to help reduce relapse, symptom severity and caregiver

burden and may be effective in addressing communication blocks identified in this study. This may be especially true of therapies that seek to educate service users and families by increasing knowledge of bipolar disorder and the associated increased risks.[48 49] Recent work examining the experiences of mental healthcare by people with bipolar disorder showed that meaningful information about bipolar disorder and its treatment (as well as psychosocial interventions and access to peer support) resulted in more positive attitudes towards care and better engagement with services.[31]

Previous qualitative work showed that people with serious mental illness struggle to be seen as competent and equal partners in care decisions,[50] and this was evident in the current work where the focus was specifically on help for suicidal behaviours. These results also support the idea that relatives and carers can feel excluded from mental healthcare.[30] As in general mental healthcare for people with bipolar disorder, care received for suicidal behaviour was viewed more positively when care was seen as collaborative and clinicians were sympathetic, open and interested in involving service users and relatives.[51]

### Methodological limitations

Total number of participants was small but sufficient to provide a novel insight into the experiences of a small and hard to reach population via rich in-depth data. Our model arose from views of service users and relatives who are both key stakeholders who provide valuable perspectives on the experiences. The relatives' cohort had a high degree of information power on the subject of service responses to suicide as all had been bereaved by suicide.[52] The service user sample was more heterogeneous with a wide range of experience of suicide thoughts, feelings and behaviours and the final model was supported by data from both sets of participants. Given the heterogeneity of the service user group, further research may be needed to ensure all experiences are accurately captured. Themes and conceptual models presented in this work do not attempt to give an exhaustive account of all experiences of accessing mental healthcare by people with suicidal behaviours in bipolar disorder. However, identification of factors considered important by participants with diverse perspectives and experiences have been described, and common difficulties faced by this high-risk group when trying to access care have been successfully identified.

### Clinical implications

Many people who experience suicidal behaviour in bipolar disorder are in close proximity to care services. While the influence of rapidly changing mood-states and impulsivity make prediction and prevention of suicidal behaviours in this group particularly difficult, the experience of people with suicidal behaviours and bipolar disorder may have a role in helping to improve the response of mental health services.[53–55] Rapid access to care is crucial, and service users and relatives need good knowledge of what services are available to them, alongside clearly signposted ways to access these services. Relatives are often in a position to observe behavioural changes that may indicate increased

or fluctuating risk, and where possible, should be kept informed of treatment/management plans. Closer involvement of family and friends may allow important information to be communicated more quickly, help build trust and supportive therapeutic relationships, and create more positive judgements of overall quality of care provided by mental health services.[56]

There may also be a role for improvement in health professionals' knowledge of bipolar disorder and the increased risk of suicide conferred by this illness. Participants' accounts consistently showed some level of underestimation of illness severity and suicide risk by clinicians, even for individuals under mental healthcare. Additional interview work with clinicians may help uncover problems in communication about suicidal behaviour in people with bipolar disorder, and help explore the perception of lower risk in people who are better able to communicate their distress and concerns. Further work with a wider range of participants, particularly those from less educated backgrounds where additional barriers may include health literacy, is needed to explore the generalisability of the themes presented as well as how themes might be translated into improved care practices.

**Contributors** NK and CC designed the study with input from all authors. CC collected all data. CC and SP analysed and interpreted the data, with input from all authors. CC drafted the manuscript. NK, SP, RM, SHJ provided further input on subsequent drafts of the manuscript.

**Funding** This article presents independent research commissioned by the National Institute for Health Research (NIHR) under its Programme Grants for Applied Research funding scheme (RP-PG-0407-10389).

**Disclaimer** The views expressed in this work are those of the authors and not those of the NHS, NIHR or Department of Health.

**Competing interests** None declared.

**Patient consent for publication** Not required.

**Ethics approval** Ethical approval for the study was obtained from the National Research Ethics Service (Ref: 10/H1003/84) and R&D approvals were granted from all trusts involved in recruitment.

**Provenance and peer review** Not commissioned; externally peer reviewed.

**Data availability statement** No data are available.

**ORCID iD**
Caroline Clements http://orcid.org/0000-0003-4735-6728

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
