## [Reviewer comments · BMJ Open]

ARTICLE DETAILS

TITLE (PROVISIONAL)	A qualitative investigation of relatives' and service users' experience of mental health care for suicidal behaviour in bipolar disorder.
AUTHORS	Clements, Caroline; Kapur, Navneet; Jones, Steven; Morriss, Richard; Peters, Sarah

VERSION 1 – REVIEW

REVIEWER	Robin E. McGee Emory University, Rollins School of Public Health, Atlanta, GA, USA
REVIEW RETURNED	15-Apr-2019

GENERAL COMMENTS	This paper contributes to the field by exploring areas for intervention to reduce suicide among people with bipolar disorder. The authors have taken care to describe the purpose, methods, and results well. Areas of improvement are noted below. Methods: 1) Justify why you needed to interview people with bipolar who self-harmed and relatives of people who died by suicide. It is not clear why the two different samples were included together for this paper.2) It may be helpful to provide an example of how the interview guide was amended based on feedback from the service user.3) The analysis portion is not well described even though some of details are provided in supplementary table 1 S1. More detail is needed in the paper. I still had questions after reviewing the table too. Did you use a codebook? What are some example codes? Did only one person code the transcripts? Did you compare and contrast responses from your two different participant groups? How did the two different participant groups factor into your analysis? How did you develop and refine your conceptual models? Results: Minor typos throughout. Ones that I spotted include: first sentence under the “merry-go-round” theme - page 6, line 48, 49 Extra period - page 8, line 15/16 Extra space - page 8, line 23/24 Unclear language - page 8, line 37 - what does “As in work on people..”, mean? Your results could be more clearly organized. I suggest highlighting the specific sub-themes in your introductory paragraph to each theme.
--

	Your results could be more specific in places too: what does “a less common but not rare” finding mean (page 8, line 42)? Can you provide more context about what you mean by that phrase? Additionally, in places where you have two-sides of an issue, more clarity is needed. Was one side more present in the data than another side? And how did participants vary in their descriptions of their experiences? For instance, in your description of gatekeeper as allies and barriers, were these experiences that all participants described? Was it that sometimes they had good experiences and sometimes they had bad experiences? Or was it that some people described gatekeepers as allies and some people described gatekeepers as barriers? In another place, how present in the data was the ‘improvement in services’ what does “some evidence” mean? (Page 12, line 48/49) Language: What do you mean by disorganized lifestyles? Page 10, line 50/51 Can you use more specific language? Conclusion: You mention that stigma did not come up as a theme from your participants. It is hard to know from your description of your methods whether this might be due to the questions you asked and focus of the interviews or because it was not a factor in their experiences. It might help to put some more context around the disclosure findings. Could stigma have played a role in not wanting to share information related to suicidal ideation? What does the literature say about not disclosing suicidal ideation? You discuss education in your conclusion, but your analysis does not examine anything about education and your demographic table does not share these details. Table 1. I don’t think the details about the method of suicide/self-harm are relevant to your analysis or needed in this paper. Please remove from table. How did time since last act play a role in your analysis? Should the participant demographics be broken out by your two participant groups? Does education-level need to be included? Supplementary Table S2 - repeat quote at the end Supplementary Table S2 - repeat quote in relatives communication Reporting Checklist Researcher Characteristics and Reflexivity is always applicable in qualitative research. You should include details about how you dealt with your position and how you were reflexive throughout the study in your methods section or in your conclusion.
--	--

REVIEWER	Barbara D’Avanzo Istituto di Ricerche Farmacologiche Mario Negri IRCCS, Milan, Italy
REVIEW RETURNED	30-Apr-2019

GENERAL COMMENTS	This is an interesting paper addressing the experiences of people with bipolar disorders who self-harmed or relatives of people with bipolar disorders who died by suicide. The issue is of paramount importance and it surely benefits from a qualitative approach. The
--

	findings are promising, but I have some quite serious concerns, listed below. 1. Interviews by telephone are not very common in qualitative studies. I agree that some people may like better to be interviewed by phone, but this surely makes a difference, which should be tackled in the Discussion. If the interviewers did not find any meaningful difference between in-person and telephone interviews (which would surprise me), they should say that.2. In the Methods section, there is no mention of the authors' position as subject of the research, and in the Reporting checklist they reported n/a with regard to the reflexivity issue: this is strange, since the authors are expert and outstanding authors in the field of suicide, and this necessarily entails positions, opinions and beliefs. I think that in this case some non-ritual thoughts about reflexivity is even more necessary. Also influences potentially deriving from more personal characteristics or political positions, or other theoretical interests should be acknowledged here. I think this aspect - the continual internal dialogue and self-evaluation of the researcher about their position and how this is related to the topic, and also the explicit acknowledgement of such position and how it can affect the study - of qualitative research is quite neglected, but it is central to qualitative investigation and can add a lot to the work, if well interpreted. So, I think that the authors should "reflect" about this in the Methods.3. Results. Whereas I am rather satisfied with the subthemes, I am not with the two main themes, "Merry-go-round of access to mental health care" and the "Routes and roadblocks in communication". They are more general definitions useful to group the subthemes together, more than a way to give them more meaning and strength. The subthemes do help understand the experience of the interviewees, whereas the two themes do not at all. They are more titles than themes. I ask the authors to try some more effort to reach a more convincing synthesis of their codes and subthemes.4. Consistently with the previous point, I think that the two models presented in the Figures need to be explained. This might solve the problems I have raised above, since a more thorough presentation of the two models could help understand their meaning. The model of Figure 1 shows the relationship between Access to care and the four factors around, which is absolutely apparent, and I would have expected a sort of representation of the relationships among the four factors, which are not at the same logical level. In Figure 2, the inner circle "Evaluation of care" corresponds to the first subtheme, but in the figure it seems to have a sort of central position (I guess because the evaluation of care is influenced by collaboration and communication among the three figures). So, it seems that there is some inconsistency between findings and models or that something is lacking in the findings presentation or in the model.5. Among the clinical implications, I would suggest to consider the need for professionals' better knowledge of bipolar disorders and suicide behavior. The accounts consistently report some underestimation of the risk of suicide by professionals. In the rationale, the authors address the evidence that people with bipolar disorder who die by suicide have been visited by a professional short before the event. They underline the finding at page 13. So, communication is surely central, and it should be supported by more professionals' skills in communication about suicide and assessing the risk.
--	---

	6. I would also consider working with professionals, in order to understand what beliefs lie under the limited ability to communicate about suicidal behaviours, the idea that people who can speak more properly can better control their emotions and suffering. This would be highly in agreement with the issue of communication among all those involved in the problem. 7. Discussion and use of references. Whereas the authors are extremely familiar with quantitative literature, it would be interesting to compare findings and conclusions with those from other qualitative investigations. For instance, they could discuss their findings in connection with those from a previous qualitative study (Owen et al, 2015), although differently focused. 8. Not many epidemiological studies are necessary to show consistency from what we know from epidemiology and the present qualitative findings. In a qualitative study, the Discussion should not be focused only on showing consistency with what is already known, but rather give clues for new perspectives. This does not mean that all qualitative studies should show something “completely new”, but the exploratory nature of qualitative study should be underlined. Minor issues.  • In the Introduction, a recent paper about experiences of people with bipolar disorders in the health care system should be cited: Vallarino et al (2019). Experiences of Mental Healthcare Reported by Individuals Diagnosed with Bipolar Disorder: An Italian Qualitative Study. Community mental health journal, 55(1), 129-136. This could also be used in the Discussion. • Is there any connection with the Owen et al (2015) study? The two studies seem to be conducted in the same area with similar recruitment procedures. If any connection exists, this should be stated. • There are some few mistakes or missing words. For instance page 14, line 13 (ref) is still there, and at the line just below maybe “of” is lacking between knowledge and bipolar. And a few more minor things throughout the manuscript. Given the importance and the relative novelty of the topic, I think that the paper needs to be improved. It can surely contribute to a better understanding of how to approach people with bipolar disorders and suicidal behavior in general, but it needs to go more in depth of the findings.
--	--

VERSION 1 – AUTHOR RESPONSE

Reviewer: 1

This paper contributes to the field by exploring areas for intervention to reduce suicide among people with bipolar disorder. The authors have taken care to describe the purpose, methods, and results well. Areas of improvement are noted below.

Author response: The authors thank the reviewer for their comments which will help to improve the manuscript. Each point is addressed below.

Methods:

1) Justify why you needed to interview people with bipolar who self-harmed and relatives of people who died by suicide. It is not clear why the two different samples were included together for this paper.

Author response: Suicidal behaviours are conceptualised as taking place on a spectrum – with minor forms of self-harm at one end and completed suicide at the other, and people with bipolar disorder are known to be at increased risk for all types of suicidal behaviour. The two types of participant therefore provide information across the full range of suicidal behaviours in bipolar disorder. Relatives/carers are commonly included in suicide research as an information source as there may be little information available from other sources, such as patient records (see for example Isometsä, 2001 & Sher, 2013). We have now expanded the justification for the inclusion of both groups in the methods section, under ‘participants’.

[Page 5, Para 3] ‘Two populations were included in order to encompass the full range of suicidal behaviour in bipolar disorder. While people with bipolar disorder who have experience of self-harm can speak to their own experiences, excluding information around those who died by suicide would neglect the experiences of a group who may have the highest need for intervention by mental health services. People bereaved by suicide are often included as informants in suicide research as they are well positioned to know details of the individual and circumstances leading to suicide unavailable from clinical records [31,32]. Although some significant events may be unknown to the bereaved person (as well as being subject to recall bias and the influence of distress on over/under emphasis of salient factors) it is recognised as a pragmatic approach.’

2) It may be helpful to provide an example of how the interview guide was amended based on feedback from the service user.

Author response: The authors thank the reviewer for this comment. The statement in question has now been replaced as it was thought to be misleading in its original form. Further comments on the amendments to the topic guide are now included in the method section as follows:

[Page 8, Para 2] ‘A first draft of the topic guide was reviewed by a service user advisory group which included people with bipolar disorder and carers. The topic guide was confirmed to be generally suitable for the purpose of the project, but the group advised that more detail be collected on past history of self-harm and interaction with services prior to the most recent episode of suicidal behaviour to give additional context to participants’ experiences. The interview was also piloted with a service user who confirmed the interview procedure and topic guide were appropriate for the research topic.’

3) The analysis portion is not well described even though some of details are provided in supplementary table 1 S1. More detail is needed in the paper. I still had questions after reviewing the table too. Did you use a codebook? What are some example codes? Did only one person code the transcripts? Did you compare and contrast responses from your two different participant groups? How did the two different participant groups factor into your analysis? How did you develop and refine your conceptual models?

Author response: The authors thank the reviewer for their constructive comments. The methods section and supplementary information has now been expanded to provide further details of the analysis. Please see Table S1 for full amendments. The method section has been expanded as follows:

[Page 8, Para 3] ‘In brief: While initial coding was conducted by one researcher (CC), transcripts were also read by another member of the research team with expertise in qualitative methods (SP) to confirm concordance of coding and ideas during analysis. Participants’ accounts were given preliminary codes which were organised within a text document. An iterative process of discussion and review between the two researchers (CC and SP) consolidated codes into potential themes (similar codes were grouped in the working document). Themes identified were summarised and presented and discussed with the extended research team to further refine final themes (more detailed information is presented in Supplementary Table S1).’

Results:

1 Minor typos throughout. Ones that I spotted include:

first sentence under the “merry-go-round” theme - page 6, line 48, 49

Extra period - page 8, line 15/16

Extra space - page 8, line 23/24

Unclear language - page 8, line 37 - what does "As in work on people..", mean?

Author response: The manuscript has now been thoroughly reviewed to correct for typographical errors.

2. Your results could be more clearly organized. I suggest highlighting the specific sub-themes in your introductory paragraph to each theme.

Author response: The manuscript has been amended in line with the reviewer's suggestion.

3. Your results could be more specific in places too: what does "a less common but not rare" finding mean (page 8, line 42)? Can you provide more context about what you mean by that phrase? Additionally, in places where you have two-sides of an issue, more clarity is needed. Was one side more present in the data than another side? And how did participants vary in their descriptions of their experiences? For instance, in your description of gatekeeper as allies and barriers, were these experiences that all participants described? Was it that sometimes they had good experiences and sometimes they had bad experiences? Or was it that some people described gatekeepers as allies and some people described gatekeepers as barriers? In another place, how present in the data was the "improvement in services" what does "some evidence" mean? (Page 12, line 48/49)

Language:

What do you mean by disorganized lifestyles? Page 10, line 50/51 Can you use more specific language?

Author response: The authors thank the reviewer for raising this issue. The manuscript has now been reviewed for clarity and additional information provided where results were unclear. Please see highlighted sections of manuscript for details.

4. Conclusion:

You mention that stigma did not come up as a theme from your participants. It is hard to know from your description of your methods whether this might be due to the questions you asked and focus of the interviews or because it was not a factor in their experiences. It might help to put some more context around the disclosure findings. Could stigma have played a role in not wanting to share information related to suicidal ideation? What does the literature say about not disclosing suicidal ideation?

You discuss education in your conclusion, but your analysis does not examine anything about education and your demographic table does not share these details.

Author response: Reference to of the possible role of stigma in relation to accessing mental health care for suicidal behaviour is now included in the results and discussion section of the manuscript. While education level was not a separate theme within the analysis, it was noticeable from the history provided by participants that level of educational tended to be high across the sample. As this may be clinically relevant (especially in relation to the underestimate of risk in people who are better able to communicate distress) the authors considered it important to include discussion of this within the work. Demographic details have been purposefully kept to a minimum in this manuscript due to the high risk of de-anonymization posed in relation to this highly specific and relatively small group. Please see highlighted sections of manuscript for details as changes are extensive.

5. Tables

Table 1. I don't think the details about the method of suicide/self-harm are relevant to your analysis or needed in this paper. Please remove from table.

How did time since last act play a role in your analysis?

Should the participant demographics be broken out by your two participant groups?

Does education-level need to be included?

Supplementary Table S2 - repeat quote at the end

Supplementary Table S3 - repeat quote in relatives communication

Author response: The tables have been amended in line with the reviewer's suggestion. Method of suicide has now been removed as has time since last suicidal act, retaining only age-group and gender. Again, the risk of de-anonymization prevents the further breakdown of these characteristics into the two participant groups. See response to query about education given in response to point 4 above.

6. Reporting Checklist

Researcher Characteristics and Reflexivity is always applicable in qualitative research. You should include details about how you dealt with your position and how you were reflexive throughout the study in your methods section or in your conclusion.

Author response: The authors acknowledge the omission of information regarding reflexivity from the manuscript, and have now amended the manuscript (and checklist) as follows:

[Page 6, Para 2] 'Throughout the planning and conduct of the study there was awareness that the authors are experts in the field of self-harm and suicide research. The authors' prior practical and theoretical knowledge of existing models of suicide and self-harm as well as the clinical management of suicidal behaviours in mental health services made it essential to ensure that participants voices were genuinely represented. The authors attempted to critically examine their work at every stage of conduct and analysis to ensure that the resulting themes and conclusions were represented in the participants' accounts. This was accomplished in two ways; a) review and amendment of the topic schedule by an independent service user group and a semi-structured approach to interview that encouraged participants to direct the conversation and raise additional factors not included in the topic guide and, b) at every stage of the analysis authors checked identified themes against manuscripts to confirm they were actually present in participants' accounts and that themes were not unduly shaped by the researchers prior positions.'

Reviewer: 2

This is an interesting paper addressing the experiences of people with bipolar disorders who self-harmed or relatives of people with bipolar disorders who died by suicide. The issue is of paramount importance and it surely benefits from a qualitative approach. The findings are promising, but I have some quite serious concerns, listed below.

Author response: The authors thank the reviewer for their very helpful comments. Details of amendments to the manuscript are provided below.

1. Interviews by telephone are not very common in qualitative studies. I agree that some people may like better to be interviewed by phone, but this surely makes a difference, which should be tackled in the Discussion. If the interviewers did not find any meaningful difference between in-person and telephone interviews (which would surprise me), they should say that.

Author response: Due to the highly specific nature of the participant group and the small total population from which to draw (although risk of suicide is high in bipolar disorder, these events are still relatively rare when taken at a population-level), the decision to use both face-to-face and telephone interview was primarily pragmatic. No meaningful differences were identified in the codes between the telephone and face-to-face interviews, and we have now added justification to the method section.

[Page 7, Para 5] 'The use of both interview methods was primarily a pragmatic decision to aid recruitment and thereby access a range of perspectives and experiences, there are potential advantages and disadvantages to using both telephone and face-to-face interviews. Interviews via telephone can add distance and detachment during a potentially sensitive interview and help people feel more comfortable disclosing personal information[34,35]; however, face-to-face methods are known to increase rapport in the interview setting and may help put participants at ease [36,37]. There was no discernible differences identified in relation to topics covered in interviews or themes identified in the analysis, between the telephone and face-to-face interviewees.'

2. In the Methods section, there is no mention of the authors' position as subject of the research, and in the Reporting checklist they reported n/a with regard to the reflexivity issue: this is strange, since the authors are expert and outstanding authors in the field of suicide, and this necessarily entails positions, opinions and beliefs. I think that in this case some non-ritual thoughts about reflexivity is even more necessary. Also influences potentially deriving from more personal characteristics or political positions, or other theoretical interests should be acknowledged here. I think this aspect - the continual internal dialogue and self-evaluation of the researcher about their position and how this is related to the topic, and also the explicit acknowledgement of such position and how it can affect the study - of qualitative research is quite neglected, but it is central to qualitative investigation and can add a lot to the work, if well interpreted. So, I think that the authors should "reflect" about this in the Methods.

Author response: The authors thank the reviewer for raising this interesting and important point. The 'methods' section has now been expanded to include a paragraph on reflexivity in the context of this work. For details please see the response to reviewer 1 (point 6) who raised the same issue.

3. Results. Whereas I am rather satisfied with the subthemes, I am not with the two main themes, "Merry-go-round of access to mental health care" and the "Routes and roadblocks in communication". They are more general definitions useful to group the subthemes together, more than a way to give them more meaning and strength. The subthemes do help understand the experience of the interviewees, whereas the two themes do not at all. They are more titles than themes. I ask the authors to try some more effort to reach a more convincing synthesis of their codes and subthemes.

Author response: The authors thank the reviewer for their comments. The analysis has been re-visited and amended to create more meaningful main themes. Please see the results section [Page 8] for details of amendments.

4. Consistently with the previous point, I think that the two models presented in the Figures need to be explained. This might solve the problems I have raised above, since a more thorough presentation of the two models could help understand their meaning. The model of Figure 1 shows the relationship between Access to care and the four factors around, which is absolutely apparent, and I would have expected a sort of representation of the relationships among the four factors, which are not at the same logical level. In Figure 2, the inner circle "Evaluation of care" corresponds to the first subtheme, but in the figure it seems to have a sort of central position (I guess because the evaluation of care is influenced by collaboration and communication among the three figures). So, it seems that there is some inconsistency between findings and models or that something is lacking in the findings presentation or in the model.

Author response: Please see response to previous point – the results and models have been reviewed and amended to provide a more meaningful account of participant's experiences. Please see the results section and figures 1 and 2 for changes.

5. Among the clinical implications, I would suggest to consider the need for professionals' better knowledge of bipolar disorders and suicide behaviour. The accounts consistently report some underestimation of the risk of suicide by professionals. In the rationale, the authors address the

evidence that people with bipolar disorder who die by suicide have been visited by a professional short before the event. They underline the finding at page 13. So, communication is surely central, and it should be supported by more professionals' skills in communication about suicide and assessing the risk.

Author response: This important point has now been added to the clinical implications section.

[Page 18, Para 4] 'There may also be a role for improvement in health professionals' knowledge of bipolar disorder and the increased risk of suicide conferred by this illness. Participants' accounts consistently showed some level of underestimation of illness severity and suicide risk by clinicians, even for individuals under mental health care. A combination of greater knowledge of bipolar disorder in general and increased communication skills in the identification of suicidal thoughts and behaviours may be beneficial.'

6. I would also consider working with professionals, in order to understand what beliefs lie under the limited ability to communicate about suicidal behaviours, the idea that people who can speak more properly can better control their emotions and suffering. This would be highly in agreement with the issue of communication among all those involved in the problem.

Author response: This important point has now been added to the clinical implications section.

[Page 19, Para 2] 'Additional interview work with clinicians may help in understanding problems in communication about suicidal behaviour in bipolar disorder, and help explore the perception of lower risk in people who are better able to communicate their distress and concerns, as found in the current work.'

7. Discussion and use of references. Whereas the authors are extremely familiar with quantitative literature, it would be interesting to compare findings and conclusions with those from other qualitative investigations. For instance, they could discuss their findings in connection with those from a previous qualitative study (Owen et al, 2015), although differently focused.

Author response: The discussion has now been amended to include reference to the work suggested. Please see highlighted sections of manuscript as changes are extensive.

8. Not many epidemiological studies are necessary to show consistency from what we know from epidemiology and the present qualitative findings. In a qualitative study, the Discussion should not be focused only on showing consistency with what is already known, but rather give clues for new perspectives. This does not mean that all qualitative studies should show something "completely new", but the exploratory nature of qualitative study should be underlined.

Author response: The discussion section has now been amended in line with the reviewer's comments. Please see highlighted sections of manuscript as changes are extensive.

Minor issues.

- In the Introduction, a recent paper about experiences of people with bipolar disorders in the health care system should be cited: Vallarino et al (2019). Experiences of Mental Healthcare Reported by Individuals Diagnosed with Bipolar Disorder: An Italian Qualitative Study. Community mental health journal, 55(1), 129-136. This could also be used in the Discussion.

Author response: this paper has now been amended to include the suggested reference. Please see highlighted sections of manuscript as changes are extensive.

- Is there any connection with the Owen et al (2015) study? The two studies seem to be conducted in the same area with similar recruitment procedures. If any connection exists, this should be stated.

Author response: There is no direct connection between the current study and the Owen et al (2015) study. The papers do however have an author in common (Prof Steven Jones –a leading researcher in bipolar disorder), and the lead authors were affiliated with the same institution at the time the work was conducted.

- There are some few mistakes or missing words. For instance page 14, line 13 (ref) is still there, and at the line just below maybe “of” is lacking between knowledge and bipolar. And a few more minor things throughout the manuscript.

Author response: The manuscript has now been reviewed in full to resolve minor errors.

VERSION 2 – REVIEW

REVIEWER	Robin McGee Emory University, Atlanta, GA, USA
REVIEW RETURNED	26-Aug-2019

GENERAL COMMENTS	Thank you for the revised manuscript. The inclusion of reflexivity and provision of more description for the results is helpful. A few remaining concerns are noted below. Introduction: To help justify your inclusion of participants who self-harmed along with relatives of people who died by suicide, it would help to provide a more explicit link for how self-harm is associated with suicide. It is not clearly stated in the introduction right now. Pg 4 Would suggest saying something like “to the best of our knowledge there has been no investigation specific to...” Methods: Thank you for making adjustments based on the reviews. The flow of the methods could use some editing to integrate the new additions into the manuscript more fully. How did you develop the models you present? More detail about this part of your analysis is warranted. In your limitations, you discuss that relatives may have different perspectives, but this is not addressed in your analysis or in your presentation of results. Results: The distinction between relatives and participants is confusing. Aren't some relatives also participants? In the results, you present the model for how the relationships between the subthemes could be interpreted, but your methods do not describe any analysis you conducted to identify these relationships.
---

REVIEWER	Barbara D'Avanzo Istituto di Ricerche Farmacologiche Mario Negri IRCCS
REVIEW RETURNED	05-Sep-2019

GENERAL COMMENTS	I appreciate the extensive work done and the improvement of the paper. Quotations are appropriate and clear. I also appreciate the
--

	changes made in response to the comments of reviewer #1. However, I have a few more comments. 1. I find the analysis much more convincing, and I like the synthesis of the two subthemes. No connections among the two themes are highlighted and hence a synthesis of the two is not achieved. This can be a limit, but I would accept it as it stands. Anyway, Figure 1 should represent the connections explained in the text, which are not understandable now. For instance, use brief sentences around the circle like in Figure 2. Also explain this better in the text (how exactly “help seeking” is linked to “gatekeepers” and “obtaining a correct diagnosis”?). 2. I still have some concerns about the reflexivity issue. The paragraph is not really clear: Who had such awareness? The researchers or the participants? Why should knowledge etc. make essential ensure...? Maybe the authors mean they know how important and challenging “being objective” is? Please clarify these sentences. I would like to be more careful about this issue, in order to not have it oversimplified. Strictly speaking, reflexivity has to do more with acknowledgement of the implications of the researchers’ positions than with the way to ensure objectivity. The checklist definition is in agreement with this: “Researchers’ characteristics that may influence the research, including personal attributes, qualifications / experience, relationship with participants, assumptions and / or presuppositions; potential or actual interaction between researchers’ characteristics and the research questions, approach, methods, results and / or transferability”. Such influences and interactions are not negative: ideally, they should rather add meanings to the findings. Many authors acknowledge how difficult this is, but essential to qualitative research (see Finlay L, Qual Health Res 2002, 12(4): 531). So, reflexivity would entail a further step in the research where the data are not only the interviewees’ voices but also the researchers positions. Nonetheless, reflexivity is also intended as the awareness that the author’s position, story, knowledge and beliefs, can unduly influence data collection and analysis and that some strategies should be taken in order to control such influence. Apparently, the authors have considered only this meaning of reflexivity. This is not surprising. In this case, I would at least keep the reflexivity issue (awareness of one’s own position and expectations and communication of this to participants) separated from the issue of rigour in having the interviewees’ opinions represented as accurately as possible. 3. The authors have used “psychiatric group(s)” in one or two instances. This language sounds rather stigmatising, and surely not suited to a journal like BMJOpen. The wording should be changed into “people with psychiatric disorders” or something equivalent. I have realised that the uploaded documents still contain tables referring to the old themes. Please update everything. Moreover, several typos and errors are still there and new ones have appeared (like it’s instead of its). I suggest to check the manuscript more carefully.
--	---

VERSION 2 – AUTHOR RESPONSE

Reviewer: 1

1. Introduction: To help justify your inclusion of participants who self-harmed along with relatives of people who died by suicide, it would help to provide a more explicit link for how self-harm is associated with suicide. It is not clearly stated in the introduction right now.

Author response: We have now included a sentence in the introduction that states the strong link that is known to exist between self-harm and suicide.

[Page 4, Para 1] "More than half of people diagnosed with bipolar disorder self-harm at least once in their life-time and self-harm is known to be a major risk factor for suicide [4,5]. Indeed, risk of suicide in people with bipolar disorder has been estimated to be 20-30 times that of the general population [3,6–9]."

2. Pg 4: Would suggest saying something like "to the best of our knowledge there has been no investigation specific to..."

Author response: The sentence has now been amended as suggested by the reviewer.

[Page 4, Para 4] "To the best of our knowledge there has been no investigation specific to accessing care and support for suicidal behaviours in this psychiatric group."

3. Methods: Thank you for making adjustments based on the reviews. The flow of the methods could use some editing to integrate the new additions into the manuscript more fully. How did you develop the models you present? More detail about this part of your analysis is warranted.

Author response: The methods section has now been reviewed and amended. Information about how the models were developed has been added – please see response to point 6 below.

4. In your limitations, you discuss that relatives may have different perspectives, but this is not addressed in your analysis or in your presentation of results.

Author response: The final analysis was not intended to compare different perspectives of the two sets of participants, but arose from analysis across the data set. We have rephrased the section in the limitations to be clearer on this point.

[Page 18, Para 2] The relatives' cohort had a high degree of information power on the subject of service responses to suicide as all had been bereaved by suicide [51]. The service user sample was more heterogeneous with a wide range of experience of suicide thoughts, feelings and behaviours and the final model was supported by data from both sets of participants. Given the heterogeneity of the service user group, further research may be needed to ensure all experiences are accurately captured.

5. Results: The distinction between relatives and participants is confusing. Aren't some relatives also participants?

Author response: The use of the terms 'relatives' and 'participants' was not intended to imply that these are separate groups. All relatives and service users interviewed were participants in the study. However, some issues raised in interviews were more pertinent to the 'relatives' group – such as the family wanting more involvement in care provided by mental health services – and the use of the terms 'relatives' and 'patients/service users' reflects this. The manuscript has been thoroughly reviewed to ensure this is clear throughout.

6. In the results, you present the model for how the relationships between the subthemes could be interpreted, but your methods do not describe any analysis you conducted to identify these relationships.

Author response: The models were created as part of the analysis and were intended to be an aid to understanding how themes might be related. The relationships were largely based on sequences of events over time as described by participants. A section has now been added to the methods to clarify this approach.

[Page 8, Para 3] "Conceptual models depicting possible relationships between subthemes were developed to aid understanding of how the different issues identified in participants' accounts may be linked. The typical progression or sequence of events over time as described by participants were noted and used to help construct the models and the relationships between subthemes. The models were developed as part of the main analysis, and were reviewed and amended by the research team as analysis progressed."

Reviewer: 2

I appreciate the extensive work done and the improvement of the paper. Quotations are appropriate and clear. I also appreciate the changes made in response to the comments of reviewer #1. However, I have a few more comments.

1. I find the analysis much more convincing, and I like the synthesis of the two subthemes. No connections among the two themes are highlighted and hence a synthesis of the two is not achieved. This can be a limit, but I would accept it as it stands. Anyway, Figure 1 should represent the connections explained in the text, which are not understandable now. For instance, use brief sentences around the circle like in Figure 2. Also explain this better in the text (how exactly "help seeking" is linked to "gatekeepers" and "obtaining a correct diagnosis"?).

Author response: The authors thank the reviewer for their comment. More detail has been added in the methods about how the models were developed (see response to reviewer 1, point 6). Relationships as depicted are one possible interpretation based on how events over time in the participants accounts. The paper has been reviewed and amended to clarify this in the methods and in the results where appropriate. However, the authors are confident that the models are suitable in their current form (these are intended to show possible relationships in a simple and easily understandable form for the non-expert reader) and therefore no changes have been made to the models.

2. I still have some concerns about the reflexivity issue. The paragraph is not really clear: Who had such awareness? The researchers or the participants? Why should knowledge etc. make essential ensure...? Maybe the authors mean they know how important and challenging "being objective" is? Please clarify these sentences.

I would like to be more careful about this issue, in order to not have it oversimplified. Strictly speaking, reflexivity has to do more with acknowledgement of the implications of the researchers' positions than with the way to ensure objectivity. The checklist definition is in agreement with this: "Researchers' characteristics that may influence the research, including personal attributes, qualifications / experience, relationship with participants, assumptions and / or presuppositions; potential or actual interaction between researchers' characteristics and the research questions, approach, methods, results and / or transferability". Such influences and interactions are not negative: ideally, they should rather add meanings to the findings. Many authors acknowledge how difficult this is, but essential to qualitative research (see Finlay L, Qual Health Res 2002, 12(4): 531). So, reflexivity would entail a further step in the research where the data are not only the interviewees' voices but also the researchers positions.

Nonetheless, reflexivity is also intended as the awareness that the author's position, story, knowledge and beliefs, can unduly influence data collection and analysis and that some strategies should be taken in order to control such influence. Apparently, the authors have considered only this meaning of reflexivity. This is not surprising. In this case, I would at least keep the reflexivity issue (awareness of one's own position and expectations and communication of this to participants) separated from the issue of rigour in having the interviewees' opinions represented as accurately as possible.

Author response: We agree with the reviewer that this section was not clear and have now rewritten it to clarify the work the researchers undertook with regards to reflexivity and understanding the subjectivity of the data generation and analysis, and the steps that were taken to manage this throughout.

[Page 8, Para 2] "Steps taken to ensure the trustworthiness of the analysis included a multidisciplinary team that comprised qualitative methodologists, clinical psychologists, psychiatrists and expert suicide and self-harm researchers. Throughout the study the team were reflexive, using individual reflexive notes and group discussions. This involved considering their own personal experiences and positions and how these may impact on the data generation, interpretation and final analysis. In doing so the researchers did not aspire to objectivity but to understand the subjectivity inherent in the process of qualitative research [41]."

3. The authors have used "psychiatric group(s)" in one or two instances. This language sounds rather stigmatising, and surely not suited to a journal like BMJOpen. The wording should be changed into "people with psychiatric disorders" or something equivalent.

Author response: The wording has now been amended as suggested by the reviewer.

4. I have realised that the uploaded documents still contain tables referring to the old themes. Please update everything. Moreover, several typos and errors are still there and new ones have appeared (like it's instead of its). I suggest to check the manuscript more carefully.

Author response: The manuscript has been fully reviewed to address minor errors.